# The Hell of Wildfires: The Impact on Wildlife and Its Conservation and the Role of the Veterinarian

**Andreia Garcês** [1,2,3,*] and **Isabel Pires** [4]

1  Instituto Politécnico de Viseu, Escola Superior Agrária de Viseu, Campus Politécnico, 3504-510 Viseu, Portugal
2  Centre for the Research and Technology of Agro-Environmental and Biological Sciences (CITAB), University of Trás-os-Montes e Alto Douro, 5001-801 Vila Real, Portugal
3  Instituto Universitario de Ciências da Saúde-CESPU (IUCS-CESPU), 4585-116 Gandra, Portugal
4  Animal and Veterinary Research Center (CECAV), University of Trás-os-Montes e Alto Douro, 5001-801 Vila Real, Portugal
*  Correspondence: andreiamvg@gmail.com

**Abstract:** Wildfires are common occurrences worldwide that can destroy vast forest areas and kill numerous animals in a few hours. Climate change, rising global temperatures, precipitation, the introduction of exotic species of plants (e.g., eucalyptus), intensive agriculture, and deforestation have increased the number of fires and their intensity and destructive power. Nearly 4% of the global land surface (30–46 million/km$^2$) is burned annually. There are many studies regarding the impact of wildfires on ecosystems, flora, domestic animals, and humans. Even though wildfires are a common and recurrent problem around the world, most of the time, they are a neglected issue, especially regarding wildlife. The information available is scarce and dispersed across several bibliographical references, and the veterinarian teams, most of the time, need to be trained to act in these situations. In this brief review, we describe different species' behavior during a wildfire, the effects on the ecosystem, and the socioeconomic impact on the region. From a veterinarian's perspective, we list the major injuries that are expected to be observed and how to proceed. In conclusion, we discuss better prevention and response measures in a wildfire scenario. This information can be used by veterinarians and all the entities involved in the prevention and combat of wildfires, and the general public has an important role in these situations.

**Keywords:** mortality; burns; conservation; wildfire; wildlife; smoke

## 1. Introduction

Wildfires are common occurrences worldwide, defined as destructive and unregulated fires in rural areas on vegetation spots (e.g., forests, savannas, grasslands) [1]. Climate change, rising global temperatures, less precipitation, the introduction of exotic species of plants that consume an excessive quantity of water (e.g., eucalyptus), heavy agriculture, and deforestation play an active role in increasing the number of fires and their intensity and destructive power [2]. In addition, environmental conditions influence the prevalence and severity of fires, which in some cases, lengthens the fire season and widens burn areas [1]. Wildfires can originate from numerous natural sources or are caused by humans [1]. Lightning, dryness, or volcanic eruptions are the leading causes of forest fires. Human-caused wildfires can be caused by accidents, such as campfires, cigarettes, burning debris, electrical shocks, equipment breakdowns, fireworks, etc. [1]. Moreover, power lines are a source of some devastating forest fires. Some of these fires can occur due to the interaction between wild animals and power lines, especially birds, such as diurnal raptors, *Corvidae*, and nocturnal raptors. In a study carried out in Spain between 2000 and 2012, of the 2788 wildfires caused by power lines, 30 were fauna-mediated. These fires destroyed 9.06 hectares of vegetation [3].

In some instances, forest fires may be caused by arson. An example occurred in Brazil and is known as "Fire Day". On 10 August 2019, numerous rural producers in the country's northern region started a joint movement to set fire to several areas of the Amazon rainforest to create pasture areas [4]. The presence of wind, organic matter, dry soil, and low humidity have accelerated the onset of a rapidly spreading wildfire that becomes difficult to control and extinguish [1].

The economic problems and losses experienced by domestic, wild, and human animals are significant consequences of fires [1]. However, in some regions, for millennia, forest fires have become a natural element of the ecosystem (e.g., Australia, North America), and wild species have a long relationship with them, at times, with benefits [1]. Savannahs, grasslands, pine forests, and Mediterranean scrubland are ecosystems that benefit from small-scale wildfires throughout the year [5–7]. Fires can help regenerate plants, increase biomass, decrease the irregularity of the habitat, increase significant diversity of food fonts, increase the production of seeds, and increase the nutritional value of plants [8]. As predators and scavengers, certain animals also benefit from forest fires [9].

Wildfires not only affect animals and the ecosystem, but they also have long-term and short-term socioeconomic impacts. Property loss (e.g., infrastructures, cars, agriculture fields, fabrics) is one of the wildfires' immediate economic impacts [10]. The loss of property is accompanied by the displacement of people from their homes, the decimation of businesses, and the substantial effects on insurers [11]. Moreover, forest fires impact tourism, as they can erase outdoor spaces that attract tourists and drive them away for years. They also negatively affect hospitality, restaurants, and other industries [12,13]. Air quality is a major risk associated with forest fires. Around eight billion tons of $CO_2$ have been emitted per year in the least the past two decades. For example, in 2015, wildfire smoke in Palangkaraya, Indonesia, increased the air quality index to 2000. It has resulted in 500,000 severe respiratory infections and an estimated 100,000 premature deaths. Water quality may also be affected, an issue that can last months or years [14]. From 2000 to 2012, the cost of forest fires in the Mediterranean region was estimated at approximately 7.6–12.4 million euros due to repairs, loss of biodiversity, and $CO_2$ emissions [3].

Nearly 4% of the global land surface (30–46 million/$km^2$) is burned annually. Portugal is a case in point among the countries most affected by forest fires in Europe. Each year, numerous forested areas are burned. Figure 1 represents the burned area in hectares from 2002 to 2009 in Portugal. According to Portugal Deforestation Rates and Statistics (GFW), 2017 was the wors year, with a total of 563,532 hectares burned [15]. In Europe, many countries have been severely affected by wildfires in the last years, particularly southern countries. For example, Greece, France, Italy, Spain, and Portugal devote nearly 2500 million euros annually to fire management [1]. In South America, particularly Brazil, a similar situation has been observed, with a vast area of forest lost to fires in the last decades [16,17]. Only in the Brazilian Pantanal, 189,440 hectares were burned in 2020 [18].

Numerous studies focus on the impacts of wildfires on ecosystems, flora, domestic animals, and humans. Nevertheless, regarding wildlife, little is still known. Currently, there is no accurate assessment of the number of animals that die each year in fires. This is due to the absence of precise numbers of wild populations in the years previous to the fire and to the difficulty of quantifying post-fire mortality because bodies are often scorched (e.g., species like amphibians and insects are too small to be counted) [9]. Furthermore, post-fire counts are rarely conducted, and it is difficult to determine if the animals died or migrated in response to the fire. Even though wildfires are a common and recurrent problem worldwide, most of the time, they are a neglected issue, especially regarding wildlife. The information is scarce, dispersed by several bibliographical references, and not easily accessible. In addition, veterinaries, biologists, NGOs, firefighters, and volunteers often need to be trained to act correctly.

In this study, the authors aim to provide a brief, easy-to-read overview of the impact of wildfires and their main consequences on wildlife. Even though it is more focused on veterinarians, it also can serve as a guideline for all the entities involved in the prevention

and combat of wildfires and the general public that has an essential role in these situations. In the first section of the review, the authors describe how the different species are expected to behave during a wildfire and the impacts on the ecosystem. Then, from a vet's point of view, we list the major injuries that are expected to be observed and how to proceed. Finally, the authors discuss the best prevention measures and how responses to forest fires can be improved.

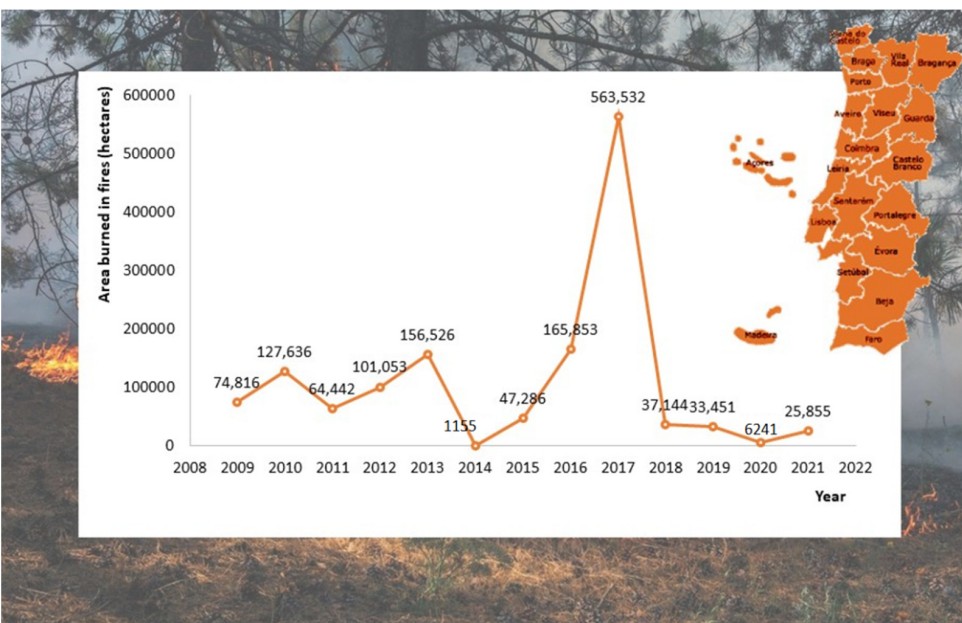

**Figure 1.** The area burned in Portugal by wildfires (in hectares) between 2008 and 2021. Adapted from [8].

Most animals can detect forest fires and identify the hazards associated with them. However, according to size and species, they have different defense mechanisms (Figure 2). Some of them run, while others can seek refuge underground or in the water. Some species may even profit from confusion and hunt small prey as they try to escape the fire [19].

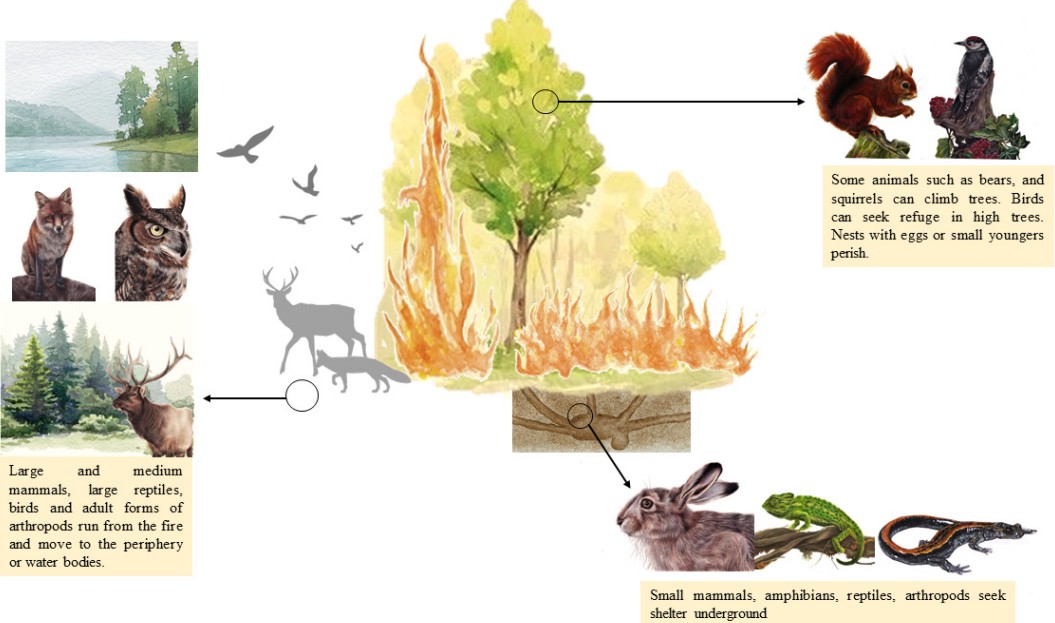

**Figure 2.** A conceptual model illustrating animal responses to wildfire (Illustrations: Andreia Gracês).

*Mammals*

The ability of mammals to survive fire will depend on their mobility, size, velocity, and duration of the fire [20]. Larger animals, such as deer, wolves, or wild boars, are highly mobile. They run from the fire and move to the periphery [21,22], and some can even swim along rivers. Some animals, such as bears and squirrels, can climb trees [9]. Small mammal species also can run from the fire in groups to clearings, road cuts, depressions, and hiking trails [22,23]. However, most small mammals, such as mice and rabbits, tend to seek refuge underground or in sheltered places within the burn, such as underground tunnels, stumps, root holes, pathways under moist forest litter, and spaces under a rock, talus, and sizeable dead wood [22].

*Birds*

Birds fly from fire zones to others, except for eggs and very young birds [8]. However, the species that fly at a lower altitude can be affected by smoke inhalation [24].

*Amphibians and Reptiles*

Larger, more mobile reptiles usually run from the fire [9,25], while smaller reptiles and amphibians have limited capacity to escape. Small lizards, turtles, and frogs seek refuge underground in holes or burrows [26]. Amphibians that live near aquatic areas can also seek shelter in water bodies [9].

*Aquatic animals*

Aquatic animals can be better protected from forest fires while living in large bodies of water. However, they have limited mobility, except for those living in running water like rivers. During a wildfire, due to the high temperatures, water can exceed the lethal limits of temperatures and vapor temperature, leading to severe heat damage [27]. This will lead to other phenomena, such as changes in pH, turbidity, accumulation of toxins [28], and excessive sedimentation affecting aquatic fauna [29–31]. Excessive sediments, such as ashes or organic debris, can squeeze or displace fish eggs in the bottom of the water bodies [32].

*Invertebrates*

Eggs, nymphs, and adult stages of arthropods may be affected due to the heat of the flames, and species with immobile life stages that live in surface litter or aboveground plant tissue usually perish [33]. Some adults burrow or fly out of flames [9,33].

## 2. Impact on Ecosystems and Wildlife

The consequences of wildfires in the ecosystem are diverse. There is a tremendous loss of fauna and flora. Habitat becomes simple and poor due to reduced diversity [34]. Shrubs and grass replace forests. Loss of forest cover results in higher temperatures in the forest soil, which can affect plant growth and animal behavior. Increased temperatures impact cavity-nesting species, such as birds and small mammals. Dead trees produce extreme temperatures in nest cavities, affecting egg incubation and the survival of heat-sensitive young birds and mammals [35].

The adverse effects of forest fires are directly related to animal injury and the destruction of nesting and breeding areas, shelters, and food sources [1,19,36]. Although some animals can be used for the occurrence of wildfires, they suffer stress. However, there are not enough studies on stress's short- and long-term effects [9]. Typically, the most affected are the slower-moving species, like turtles, badgers, and elderly and very young animals who are unable to escape [19]. Moreover, as wildfires often occur in late spring or summer, stress also delays the recovery and reproduction of the population [9].

In 2020, 78% of the total area of the jaguar territory in Pantanal (Brazil) was burned. The high number of fires that year and prior years had a negative impact on jaguar populations. The main effects observed were temporary displacements, which made it difficult to find new suitable areas, increased territorial conflicts, and decreased survival and reproductive success. In addition, many animals have suffered from starvation, dehydration, and

death, further contributing to the decline of the already threatened population of jaguars in Pantanal [37]. The animals that abandoned the area to escape the fire may return, depending on how much the fire altered the habitat structure and food supply [8]. Non-burrowing mammals, reptiles, and birds may return within hours or days [8]. After a fire, when animals return to the burned area, they inspect the environment to determine settlement options [9]. Although some animals can return to take advantage of the new habitat and adapt their diet and behavior, others cannot survive and migrate to other areas. If the habitat does not provide the structure or food necessary for survival and reproduction, they move to fire-free areas, unburnt islands, or nearby unburnt vegetation [9,38]. For example, large mammals such as deer or moose depend on a significant amount of vegetation for their diet, bedding, shelter, and thermal protection [9]. As a result, many animals die after the fire from starvation. Other animals can wander into urban and suburban areas in search of new habitats where they come into contact with humans [19].

Animals are not equally affected by stress factors following a fire. The magnitude and duration of the fire, type of vegetation, climate, smoke, and water bodies are some factors that will determine the impact on animals. Another aspect to consider is associated with the animal, such as species, physical condition, age, physiological status, response to stress, availability of food or water, or injuries (e.g., burns or smoke inhalation). Therefore, it is necessary to account for various nuances in animals and fireplaces that determine the susceptibility to illness after a fire [36]. The migration of animals to new regions is associated with pathogens (e.g., viruses, parasites) spreading or with the acquisition of new pathogens in the newly explored areas [39]. Increased contact of wildlife with people and domestic animals also increases the risk of exposure to and transmission of diseases with zoonotic potential [19]. In addition, fires can also favor the occurrence of arboviral infections. For instance, in Brazil, studies have linked fires to outbreaks of diseases, such as dengue fever, Zika virus, chikungunya, and yellow fever [39].

Critical soil biological processes alternate after the fire due to increased light, temperature, and wind from unprotected soil. Examples are humidity reduction, loss of nitrogen and carbon to the atmosphere deposit of charcoal and ash, and other physicochemical alterations (bacterial and fungal activity and population changes) in soil [40]. This results in increased canopy fracture, higher tree fall rates, differentiation of plant diversity, and downward displacement of vertical stratification of foliage density. As a result, housing and food are reduced, resulting in a shift in wildlife distribution [41]. In addition, the lack of coverage makes small species, such as mice, amphibians, lizards, and insects, more visually exposed and easily targeted by predators [41,42].

The Watershed's morphology in the long term is also affected by fires [43]. Post-fire sediments can provide new resources for aquatic animals or become a source of pollution in their habitat [44]. Some species present accelerated growth rates after recolonizing post-fire rivers [45]. On the other hand, aquatic fauna can perish due to variations in water pH, turbidity, and toxins from the post-fire sediments. Shellfish mortality is also reported in sea costs where waters full of post-fire debris flow [9].

However, some ecosystems have adapted to and benefitted from small-scale forest fires. Ash is a natural fertilizer for the soil and contributes to the growth of new seeds in these areas. Fires eliminate dead and sick plants and do not let overgrown forests occur, allowing more sunlight to reach the ground and healthier plants to grow. It allows for greater plant diversity, helping to increase ecosystem resilience by creating an island with different microhabitats [1,36]. Some animal species benefit from fires regarding foraging and nesting behaviors. The deadwood on the ground can provide food, shelter, and a cavity nest to some species. Invertebrates, small mammals, birds, amphibians, and reptiles pursue protection and cover in this down wood [8]. Predators and scavengers are often attracted to burns because their food is more abundant or exposed than on unburned sites. Necrophagous species, such as vultures, can benefit from the number of carcasses available after a fire. For example, woodpeckers are especially attracted to burning areas due to the high number of beetles and other invertebrates present in dead wood [8].

### 3. The Role of the Veterinarian: Treatment and Recovery of Burned Wildlife

Not all animals can escape from the flames. They can be burned, affected by smoke, dehydration, heat exhaustion, or suffer traumatic injuries when running the flames. Some die burned or due to smoke inhalation (Figure 3).

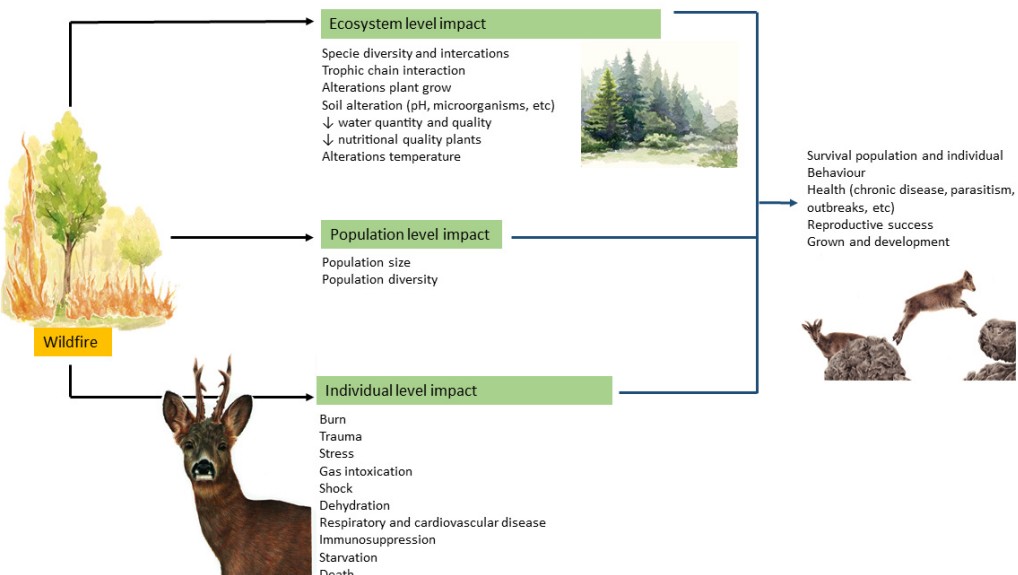

**Figure 3.** A conceptual model illustrating the different impacts of wildfire on the ecosystem, population, and individuals (Illustrations: Andreia Garcês).

Vets, biologists, NGOs, firefighters, and volunteers play a critical role in the first aid and recovery of these animals. Animals rescued from wildfires should be examined by a veterinarian to receive proper treatment and rehabilitation. The veterinarian will evaluate burns, damages sustained from smoke intoxication, traumatic injuries, pre-existing diseases, dehydration and shock level, and stress [9]. Veterinarians also have a critical role to play in informing society about actions that can further help or harm wildlife during fires and how to find victims [9].

The management of fire-affected animals shall ensure the overall assessment and clinical support of qualified clinicians. Once the animals arrive at the veterinary hospital/wildlife rehabilitation center, a complete body examination and vital signs should be collected. At this stage, it is essential to decide if animals can recover from their injuries or if euthanasia should be provided. Shock and dehydration should be treated. The animal must stay warm (24–30 °C) in a dark and quiet environment, away from human circulation and noise [19,36]. Animals often experience hypothermia due to loss of fluids, blood, or ineffective cooling attempts. The victim of the wildfire is highly susceptible to being found dehydrated. The type and amount of fluid provided to the animal will depend on its condition. When animals are conscious, it is important to provide water and encourage drinking. Always leave fresh water available. Sometimes animals are too disorientated or sore to move and may not drink immediately. Fluids can also be given subcutaneously or intravenously if the animals are unconscious or very debilitated. Ideally, a minimum of 10 to 20% dehydration should be treated for a minimum of three days. Some animals will need intensive care and ongoing treatment over a long period of time. This is a very time-consuming process. Sometimes, animals cannot be released into the wild because they lose their natural behavior during captivity. When animals change to moderate-intensity care, they should be kept in small groups and finish their rehabilitation in wide enclosures where individuals can express their natural behaviors and develop strength before being released into the wild [9].

The primary lesions presented by the victims of wildfires are burns, smoke intoxication, and acute heat stress response.

### 3.1. Burns

Burns on the face and limbs are the most common lesion in wild animals' recovery from wildfires. Some animals can escape the fire with only a few feathers or fur burns. Others have deep burns on the skin. When an animal presents burns, the first thing to consider is the burn's depth, extent, and location [46–48]. Feathers and fur may hide the appropriate extension of burns and should be set aside for accurate examination of the skin [46–48].

Burns can be classified as superficial, partial, and complete thickness according to depth (Table 1). Superficial burns are rare in wildlife since the fur, feathers, and scales provide some protection [46,49]. The most common burns are partial and complete thickness (Figure 4). Bird skin does not blister as prominently as mammalian skin as it lacks collagen. It can make it difficult to classify the type of burn [46]. Therefore, when accessing burns, it is essential to assess their severity and the welfare of the animals. If it is not possible to rehabilitate, the animal's suffering should not be prolonged. In general, burns extension inferior to 10% of the body surface has a reasonable prognosis, 10–15% of burned corporal areas have a poor prognosis. More than 20% of the body burned is irreversible, and swift euthanasia should be provided to the animal [46,49].

**Table 1.** Classification of burns in animals by their depth.

| Degree of Burn | | Skin Layer Affected | Macroscopic Exam | Pain | Prognosis |
|---|---|---|---|---|---|
| Superficial burns or first-degree burns | | Epidermis | Erythema, reddening, swelling, no blister | Present | 2 weeks without scarring |
| Partial thickness burns or second-degree burns | Superficial | Upper dermis | Diffuse erythematous or mottled, moist surface, reddening, swelling, blister, and weeps, blanching on pressure | Present | 2–3 weeks without scarring |
| | Deep | Lower dermis | Blotchy with red or white areas, dissuade erythematous, moist, dermal oedema, blister, and weeps, no blanching on pressure | Some | 2–8 weeks to heal with possible scarring, risk of infection |
| Total thickness or third degree | | Subcutaneous structures | Greyish or with necrosis, dry, blackened | Absent | >8 weeks, small areas heal with a scar, surgery, risk of infection |

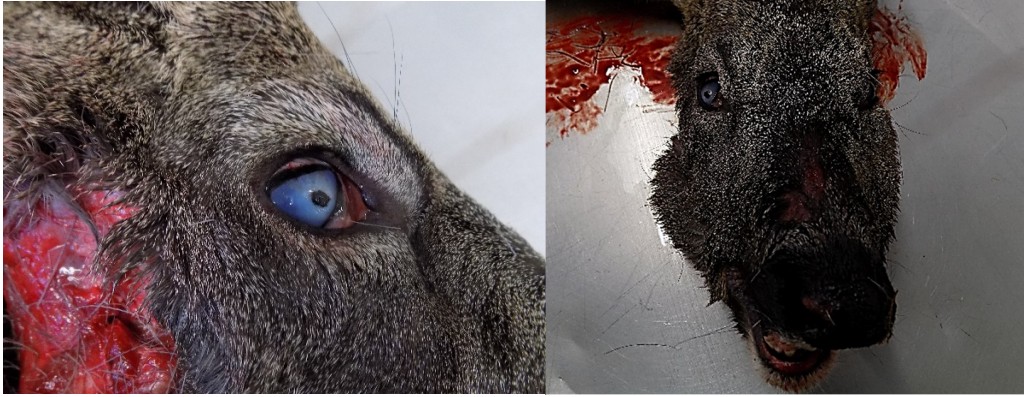

**Figure 4.** Ulceration of the eye due to high temperatures and partial thickness burns in the head of a roe deer (*Capreolus capreolus*) (Photo: Isabel Pires).

Several methods can be used to calculate the surface area burned in an animal body as a percentage of the total body surface area. One of the most straightforward methods is based on Wallace's "rule of nine" used in humans, which divides the body into regions that are multiples of 9% of the total body surface area. In animals, the head and neck are counted as one "nine" or 9%, each forelimb 9%, each hind limb two "nines" or 18%, and

the dorsal and ventral halves of the trunk 18% each (Figure 5). However, this method has limitations since there is significant variability between animal body shapes. For example, surface areas have not been determined for macropods due to the great diversity of tail lengths. Another method used is the "Resuscitation Burn Card," which gives a more precise measure (Figure 6) [47,49].

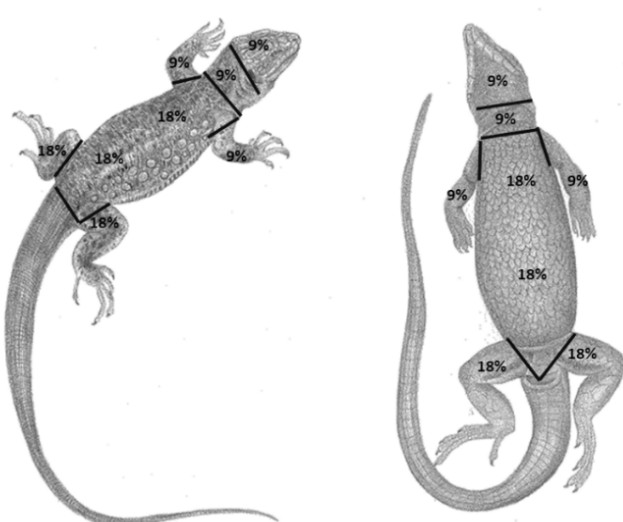

**Figure 5.** Estimating the extent of the burned area according to the "rule of 9" schematically depicted in a lizard (Illustration: Andreia Garcês).

### Veterinary Burn Card

To calculate the burn percentage of the total body surface area (% TBSA):

(1) Measure the burn with this burn card. How many burn cards are needed to cover the burn completely?

(2) Weigh the patient. Convert the weight to surface area in m² using the conversion table on the back of this burn card.

(3) Apply these numbers to the following formula:

$$\% \text{ TBSA burn} = \frac{\text{number of cards} \times 0.45}{m^2}$$

**Example:** 21 kg dog; burn size = 15.5 burn cards

$$\text{Burn area} = \frac{15.5 \times 0.45}{0.76} = 9.2\% \text{ TBSA}$$

**Weight to Surface Area**
**CONVERSION TABLES**

| | Dogs | | | | | | Cats | | |
|---|---|---|---|---|---|---|---|---|---|
| Kg | m² | Kg | m² | Kg | m² | Kg | m² | Kg | m² |
| 1 | 0.10 | 20 | 0.74 | 46 | 1.28 | 1.0 | 0.100 | 6.0 | 0.330 |
| 2 | 0.15 | 21 | 0.76 | 48 | 1.32 | 1.5 | 0.131 | 6.5 | 0.348 |
| 3 | 0.20 | 22 | 0.78 | 50 | 1.36 | 2.0 | 0.159 | 7.0 | 0.366 |
| 4 | 0.25 | 23 | 0.81 | 52 | 1.41 | 2.5 | 0.184 | 7.5 | 0.383 |
| 5 | 0.29 | 24 | 0.83 | 54 | 1.44 | 3.0 | 0.208 | 8.0 | 0.400 |
| 6 | 0.33 | 25 | 0.85 | 56 | 1.48 | 3.5 | 0.231 | 8.5 | 0.416 |
| 7 | 0.36 | 26 | 0.88 | 58 | 1.51 | 4.0 | 0.252 | 9.0 | 0.432 |
| 8 | 0.40 | 27 | 0.90 | 60 | 1.55 | 4.5 | 0.273 | 9.5 | 0.449 |
| 9 | 0.43 | 28 | 0.92 | 62 | 1.58 | 5.0 | 0.292 | 10.0 | 0.464 |
| 10 | 0.46 | 29 | 0.94 | 64 | 1.62 | 5.5 | 0.311 | | |
| 11 | 0.49 | 30 | 0.96 | 66 | 1.65 | | | | |
| 12 | 0.52 | 32 | 1.01 | 68 | 1.68 | | | | |
| 13 | 0.55 | 34 | 1.05 | 70 | 1.72 | | | | |
| 14 | 0.58 | 36 | 1.09 | 72 | 1.75 | | | | |
| 15 | 0.60 | 38 | 1.13 | 74 | 1.78 | | | | |
| 16 | 0.63 | 40 | 1.17 | 76 | 1.81 | | | | |
| 17 | 0.66 | 42 | 1.21 | 78 | 1.84 | | | | |
| 18 | 0.69 | 44 | 1.25 | 80 | 1.88 | | | | |
| 19 | 0.71 | | | | | | | | |

**Figure 6.** The "Veterinary Burn Card" has the exact dimensions of a plastic credit card and covers exactly 45 cm². Both sides of the card have been enlarged here to facilitate readability.

For wildlife, whose release depends on a functional body, some locations of burns may influence rehabilitation. Damage near joints where scar tissue restricts the movement of limbs or digits significantly affects tree-dwelling animals such as squirrels or genets. Moreover, lesions in structures such as eyelids and mouth can affect the ability to feed or sight. Blind animals cannot survive in the wild. Nails can be used to climb trees, eat, predate, dig, groom, fight, and do other essential activities. The animal may face a lost nail on a hand, but the loss of several nails can affect its survival [9,46]. With skin damage, fluid and electrolytes are lost from the body, and the animals become more susceptible to secondary infections by bacteria or fungi. The movement becomes painful, and blood loss can occur as the tissues are fragile and unable to cope with trauma [9,46].

Treatment includes flushing the burn with tepid water of 0.9% saline for 10 min to cool down the area and clean debris, such as dirt and plant material, trimming off singed fur/feathers with scissors or clippers so the skin can be examined, and trimming away any flaps of dead skin [9,46]. A 1% Iodine solution can be used to clean the area. Moist cotton buds can clean the nostrils and bathe the eyes with saline solution. The burns

can be treated with ointments such as silver sulphadiazine and chlorhexidine bandages. Antibiotics are required for a minimum of seven days, usually for two weeks, while the necrotic tissue is debrided. Analgesics and anxiolytics should be given to minimize pain and stress. Administration of vitamins is recommended. Vitamin C reduces healing time, vitamin E is an antioxidant that benefits burns, and vitamin A helps to produce healthy skin [9,46].

### 3.2. Smoke Asphyxiation

Terrestrial and aquatic are vulnerable to the inhalation of airborne toxins in the smoke (e.g., carbon monoxide (CO), hydrogen cyanide (HCN), and delicate particulate matter (PM)). A large number of animals die from smoke-induced asphyxiation [50]. Animals can suffer from carbon monoxide poisoning (mostly fatal), thermal and chemical burns in the respiratory tract, and short- and long-term respiratory diseases [47]. Furthermore, when the lungs are filled with smoke, animals cannot maintain their body temperature by evaporative cooling and can overheat [51]. Symptoms of smoke inhalation include dyspnea, tachypnoea, wheezing, polypnea, coughing, foaming at the nostrils, and tachycardia [52–54]. When thermal and chemical lesion occurs leads to fluid accumulation in the lungs (edema) [55] (Figure 7). Due to the low exchange of gases in the lungs, animals rapidly enter hypoxemia and acidosis [49]. Long-term exposure to wildfire smoke can alter or weaken the animal's immune response. They became more vulnerable to respiratory infections (e.g., pneumonia) and chronic heart disease [49,56]. Wildfire smoke also can alter animal behavior (e.g., movement, vocalization), reduce growth rates and reproductive success [16].

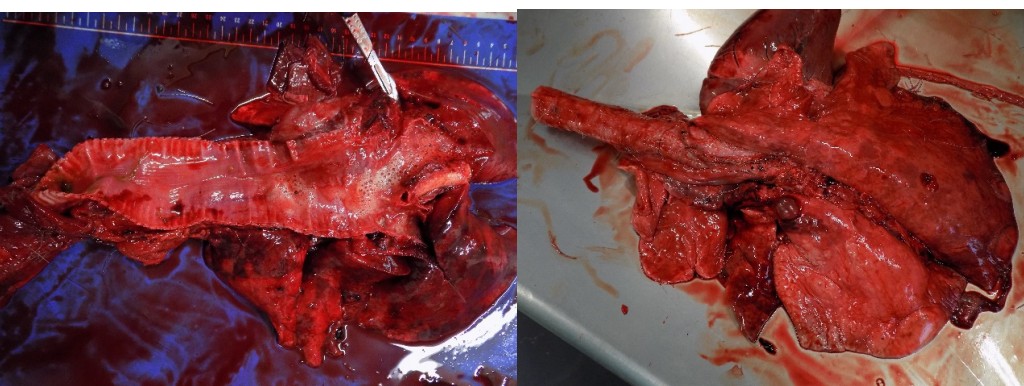

**Figure 7.** Hemorrhage, pulmonary oedema, emphysematous, and atelectatic areas in a roe deer (*Capreolus capreolus*) that inhaled smoke from a wildfire (Photo: Isabel Pires).

Treatment includes oxygen supplementation within the first 12 h of injury to help treat carboxy hemoglobinemia. Mechanical ventilation may be required to maintain oxygen saturation. Intravenous fluids can be administered to maintain average heart output and tissue infusion. Nebulization of saline and coupage might enable clearance of the respiratory secretions. In addition, it is necessary to monitor the development of secondary complications such as bacterial pneumonia and ARDS. Antibiotics, anti-inflammatory drugs, or analgesics may be required [57].

### 3.3. The Acute Heat Stress Response

Wildfires can reach temperatures higher than 63 °C. These temperatures are lethal to small animals, and it is reasonable to assume the threshold does not differ significantly for larger animals [58]. These high temperatures led to heat stress in the animals. Associated clinical signs are behavioral impairment, disorientation, hyperventilation, and discoordination [24]. Heat stress causes a decrease in food intake, hormonal and metabolic changes, tissue stress tachypnoea, an increase in skin temperature, a decrease in fertility, neurologic

conditions, and behavior change [9,59]. Nutritional supplementation (e.g., olive oil) and water during prolonged dry periods and fires can help reduce heat stress's impact [60].

*3.4. Traumatic Injuries*

When animals try to escape the flames, they become disoriented, and sometimes they can suffer accidents, such as being run over by a car, falling in holes, or crashing against structures, such as walls or trees [9]. Animals can present soft tissue and skeletal injuries, mainly affecting the extremities (e.g., fractures, internal bleeding, bruises) (Figure 8). The severity of the injury must be assessed, and when it cannot be rehabilitated, the animal must be euthanized [9]. Some birds can suffer electrocution when trying to escape [61].

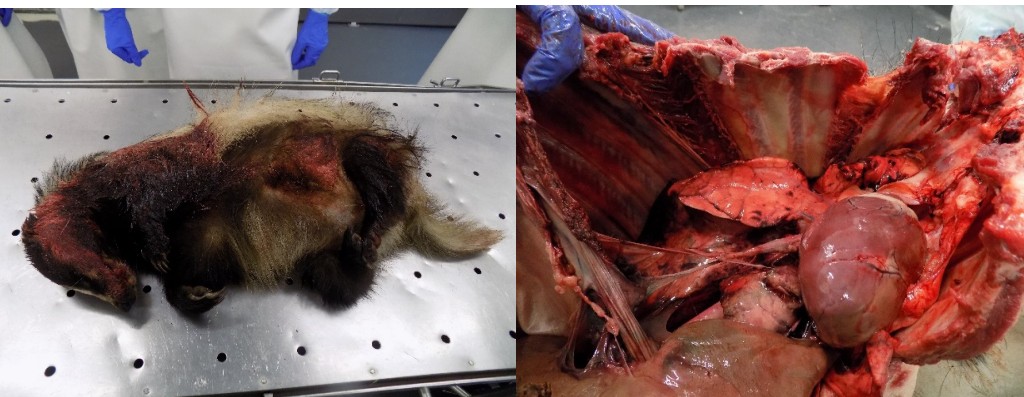

**Figure 8.** Badger (*Melus melus*) run over by a motor vehicle with multiple soft tissue and skeletal injuries. Broken ribs and rupture of intercostal muscle in a wild boar (*Sus scrofa*) that collided with a motor vehicle (Photo: Isabel Pires).

## 4. Conclusions and Remarks for the Future

There is no doubt that wildfires affect wildlife, in most cases, negatively. Over the last few decades, fires have become more widespread and destructive, and animal species cannot adapt. This is even more dangerous for the species near extinction. Threatened species populations are already so small that a fire can kill the entire population (especially animals with low mobility, such as amphibians). In addition, a fire could eradicate their habitat, leaving them to starve or unable to reproduce because their breeding sites have been damaged along with this season's youth. Veterinarians should be aware of how animals behave during forest fires and what lesions are most common in different groups of animals. This information could be essential to provide a faster and more efficient response to this catastrophe. Not only veterinarians but the public must be trained and informed about how to act. Even though forest fire is a common and recurrent problem worldwide, it is often neglected, especially regarding wildlife.

It is important to inform the public and entities involved in extinguishing the fire (e.g., firefighters) how to act in the presence of an animal victim of wildfire. For example, they should not disturb the animals that escape fires, keep pets on a leash or inside the house to avoid conflicts, supply food and create water bodies for them without making any dependence on humans, capture any animal that presents injuries and forward them to formal institutions with veterinarian assistance. In the future, further studies on the quantification of mortality and the impact of stress on the welfare of populations will be required to improve fire responses. These studies should use new software tools and traditional census methodologies to improve knowledge [9].

Wildfire management involves more measures than treating, rehabilitating, and providing a habitat for animals to return. After the fires are essential to allow the regeneration of the forests with the restriction of human access to those areas, increase soil microbial activities to reduce soil erosion, promote replantation of endemic fauna, the use of eco-friendly fuel to reduce the temperature, revegetation with new trees in the burned areas and

surrounding forest land to ensure that wildlife has access to food and shelter [1,36]. It is imperative to restore the habitat to its original status (or the closest possible), as many species have evolved to live in certain conditions. If a particular habitat disappears completely, a population can become extinct.

Unfortunately, the wildfire problem does not have a specific solution for the time being. We can try to improve response mechanisms, but the best option at this point is prevention. The combination of social governance, public and personal land management, suppression efficiency, and unique arrangement is part of the solution [1]. We should not wait for a wildfire to occur and face the consequences but rather develop pre-disaster efforts and well-organized protocols to avoid them [9]. Proper land regulation and permanent vigilance are some of the essential measures. Some measures of fire prevention are increasing forest density with endemic flora, reducing the number of non-endemic locations, creating wildlife sanctuaries to protect endangered species, and vigilance of green areas with forest guards aided by new technologies such as drones, centralized public telephone numbers and phone apps that can facilitate interventions when a fire is spotted [1]. Areas that contain endangered species must be declared protected. A large part of wildfire occurs due to human carelessness. Therefore, it is essential to inform the public and raise awareness regarding the risks of producing small fires (e.g., burns in agriculture, campfires) during dry seasons that could spread with the help of wind and cause a wildfire [1,8,9].

The future of wildfires requires better preventive and response measures. Veterinary teams should be better trained for these situations. It is also important to inform the general public about possible animal injuries and involve them in preventive measures.

**Author Contributions:** Conceptualization, A.G. and I.P.; methodology, A.G. and I.P.; software, A.G. and I.P.; validation, A.G. and I.P.; formal analysis, A.G. and I.P.; investigation, A.G. and I.P.; resources, A.G. and I.P.; data curation, A.G. and I.P.; writing—original draft preparation, A.G. and I.P.; writing—review and editing, A.G. and I.P.; visualization, A.G. and I.P.; supervision, A.G. and I.P.; project administration, A.G. and I.P.; funding acquisition, A.G. and I.P. All authors have read and agreed to the published version of the manuscript.

**Funding:** The participation of Pires I was supported by the projects UIDB/CVT/00772/2020 and LA/P/0059/2020, funded by the Portuguese Foundation for Science and Technology (FCT). (Project UIDB/CVT/0772/2020). The participation of Garcês A. was supported by National Funds by FCT-Portuguese Foundation for Science and Technology, under the project UIDB/04033/2020.

**Institutional Review Board Statement:** Not applicable.

**Informed Consent Statement:** Not applicable.

**Data Availability Statement:** Not applicable.

**Conflicts of Interest:** The authors declare no conflict of interest.

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
