# Peer review of "The Hell of Wildfires: The Impact on Wildlife and Its Conservation and the Role of the Veterinarian"

_conservation, doi:10.3390/conservation3010009_

Round 1

Reviewer 1 Report

In the first place, these authors are to be congratulated for producing a paper that is clear and informative. 

However, I cannot recommend this manuscript for publication. 

I offer two reasons:

1.  It is disjointed.  Structurally, this paper covers two topics.  The first is a review of what a fire does ecologically.  The second is a review of veterinary procedures to be used to treat burned animals.  These two topics are simply appended together without any justification. 

2.  It is unoriginal.  The descriptions that these authors provide are well-referenced.  But that is all that is here – a series of descriptions that are fully referenced.  In that sense, this manuscript could be described as a review paper.  However, even within that genre this one offers nothing new.  The informational content is readily available elsewhere, and no original points of view or analyses are provided.   

In summary, it is not so much that I am finding specific faults that need to be corrected.  The information provided is clear.  However, I can find nothing new, and hence no reason to recommend it for publication. 

Respectfully submitted. 

Author Response

Author answer: Thank you for the comment. This review does not bring new information it is true but compiles information that was started by many references. In the beginning, this manuscript has designed to be more directed to veterinarians, but the authors thought there has important to introduce what fires are due to the ecology to better understand how animals are admitted and why the type of injuries that they carry and why they have it. We try to improve a little the structure and the title.

Reviewer 2 Report

Nice summary, but without clear direction or purpose. Based on the abstract, I thought this paper was a general summary of the ecology literature on wildlife responses to wildfire. However, mid-way through the emphasis shifted to animal care, then concluded with a listing of actions related to managing the fire and wildlife issue. I found the trajectory of the paper to be confusing.

The paper was not guided by any questions or hypotheses. It did not conduct a systematic review to address a particular topic. The objectives of the paper could be more explicitly stated i.e., it is unclear why the authors wrote this manuscript and what a reader should gain from reading it. I strongly recommend adding a more explicit objectives statement to the abstract and introduction that clearly conveys what the main emphasis of the article is, and why it is important i.e., what will a reader learn from this synthesis that they will not find elsewhere.

The manuscript was fairly quick and easy to read.

Specific comments:

Grammar edit: 52

62-65: Another reason for not knowing how many animals die in wildfire, is that few such post-fire counts are likely attempted. It is also difficult to determine if animals emigrated in response to fire or died.

66-67: Suggest stating the objective here.

Section 2 seems like a preamble to section 3, with very general descriptions of responses that are good basic background. Because it is basic, it seems strange to have an entire section devoted to this content. Suggest you retain it in the introduction or just as a figure caption.

Some phrasing (e.g., 142-143) make it seems like all animals are subject to the same stressors after fire, but there is likely considerable nuance in which animals and locations become more susceptible to disease after fire. Suggest reviewing the document for instances where statements may be over-generalized and soften e.g., ‘can’ become susceptible to disease.

179- What does ‘die carbonized’ mean?

184 – abrupt switch from conceptual and general impact of fire on wildlife to the role of veterinary care.  Before this point, I thought this was a general ecology paper. An earlier objectives statement that states what you are trying to summarize with respect to veterinary care could help reduce the transition shock. The abstract mentions something vague about recovery care and prevention measures. A more explicit statement on the main topic of the paper would help cue to reader to the emphasis on post-fire animal rehabilitation and veterinary care of fire-affected animals.

The specific mention of eucalyptus in the introduction and conclusion is also confusing for a topic that is otherwise communicated quite generally. 335-343 seems a list of possible responses to fire problems but the purpose of this list and background material is unclear.

Author Response

Nice summary, but without clear direction or purpose. Based on the abstract, I thought this paper was a general summary of the ecology literature on wildlife responses to wildfire. However, mid-way through the emphasis shifted to animal care, then concluded with a listing of actions related to managing the fire and wildlife issue. I found the trajectory of the paper to be confusing.

The paper was not guided by any questions or hypotheses. It did not conduct a systematic review to address a particular topic. The objectives of the paper could be more explicitly stated i.e., it is unclear why the authors wrote this manuscript and what a reader should gain from reading it. I strongly recommend adding a more explicit objectives statement to the abstract and introduction that clearly conveys what the main emphasis of the article is, and why it is important i.e., what will a reader learn from this synthesis that they will not find elsewhere.

The manuscript was fairly quick and easy to read.

Author answer: Thank you for the suggestion, the absytract and introdution were improved.

Specific comments:

Grammar edit: 52

Author answer: The sentece was corrected to “Figure 1representes the burned area in hectares from 2002 to 2009 in Portugal. The year of 2017 was the worst year according to Portugal Deforestation Rates & Statistics (GFW), with a total of 563532 burned hectares [8].”

62-65: Another reason for not knowing how many animals die in wildfire, is that few such post-fire counts are likely attempted. It is also difficult to determine if animals emigrated in response to fire or died.

Author answer: This information was added to the text. “Additionally, post-fire counts very rarely are performed and there is a difficulty into determine if animals emigrated in response to fire or died.”

66-67: Suggest stating the objective here.

Author answer:

Section 2 seems like a preamble to section 3, with very general descriptions of responses that are good basic background. Because it is basic, it seems strange to have an entire section devoted to this content. Suggest you retain it in the introduction or just as a figure caption.

Author answer: The authors choose to add section 2 to the introdution.

Some phrasing (e.g., 142-143) make it seems like all animals are subject to the same stressors after fire, but there is likely considerable nuance in which animals and locations become more susceptible to disease after fire. Suggest reviewing the document for instances where statements may be over-generalized and soften e.g., ‘can’ become susceptible to disease.

Author answer: this section was rewritten. New sentence “Animals are not subject to the same stressors after a fire. The magnitude of the fire, type of vegetation, climate, duration of the fire, presence of smoke, and presence of water bodies, are some of the factors that will determine the impact on animals. Another aspect to have in consideration is associated with the animal as specie, physi-cal condition, age, physiological status, response to stress, availability of food or water, or injuries (e.g burns or smoke inhalation). It is necessary to consider diverse nuances in which animals and locations become more susceptible to disease after a fire.”

179- What does ‘die carbonized’ mean?

Author answer: this is regarding the animal when suffer a extrem therm lesion due to the fire the tissue is carbonized. The authors choose to change to “die burned”, since there is not not absolutely sure if it is the right term in english. 

184 – abrupt switch from conceptual and general impact of fire on wildlife to the role of veterinary care.  Before this point, I thought this was a general ecology paper. An earlier objectives statement that states what you are trying to summarize with respect to veterinary care could help reduce the transition shock. The abstract mentions something vague about recovery care and prevention measures. A more explicit statement on the main topic of the paper would help cue to reader to the emphasis on post-fire animal rehabilitation and veterinary care of fire-affected animals.

Author answer: as refered before this information was added in the introdution (objectives) and abstract.

The specific mention of eucalyptus in the introduction and conclusion is also confusing for a topic that is otherwise communicated quite generally. 335-343 seems a list of possible responses to fire problems but the purpose of this list and background material is unclear.

Author answer: the part regarding eucalyptus was deleted and the conclusion section improved.

Reviewer 3 Report

In this review, Garcês and Pires discuss the impacts of wildfires on wildlife and its conservation. This is a neglected issue, which makes this article indeed important. The text is interesting and informative. However, I have some points that I would ask the authors to improve:

- Lines 27-35: Please, Mention the "fire day" (Dia do Fogo; see https://www.bbc.com/portuguese/brasil-49453037) that happened recently in Brazil. This is an interesting example to illustrate that many fires are intentional and criminal.

- Lines 50-57: Please, include data for Brazil which is facing increasing problems with fires in the Amazon, Cerrado, Atlantic Forest and Pantanal biomes. For example, see these articles: da Silva et al. (2021) DOI: 10.1016/j.jenvman.2021.112189; Pletsch et al. (2021) DOI: 10.1590/0001-3765202120210077; Abreu et al. (2022) DOI: 10.1016/j.scitotenv.2022.157138

- Line 104: Correct the reference (Kozlowski 1974) formatting.

- Section “3. Impact on ecosystems and wildlife”: The inclusion of these articles in this section can be interesting: de Barros et al. (2022) DOI: 10.1038/s42003-022-03937-1; Brito et al. (2017) DOI: 10.1007/s11356-017-9578-0

- Lines 142-147: Please see/cite the discussion/section "Fires and other drivers of unusual movement pattern of animals" on this paper: Ellwanger et al. (2022) DOI: 10.1590/0001-3765202220211530

- Lines 184-489: Also mention the role of biologists, NGOs, firefighters, and volunteers in fighting wildfires and rescuing animals.

- Figures: It is important to mention in the legend of the figures the source of the images used (it is not clear which images belong to the authors or which were extracted from other sources).

- Lines 279-208: Check the use of punctuation and commas in this sentence.

- Line 284: Correct the references (Venn-Watson et al 2013Black et al 2017) formatting style.

Finally, I would like to congratulate the authors for this article dealing with an important but neglected topic.

Author Response

In this review, Garcês and Pires discuss the impacts of wildfires on wildlife and its conservation. This is a neglected issue, which makes this article indeed important. The text is interesting and informative. However, I have some points that I would ask the authors to improve:

- Lines 27-35: Please, Mention the "fire day" (Dia do Fogo; see https://www.bbc.com/portuguese/brasil-49453037) that happened recently in Brazil. This is an interesting example to illustrate that many fires are intentional and criminal.

Author answer: thank you for the suggestion. This information was added to the text.

- Lines 50-57: Please, include data for Brazil which is facing increasing problems with fires in the Amazon, Cerrado, Atlantic Forest and Pantanal biomes. For example, see these articles: da Silva et al. (2021) DOI: 10.1016/j.jenvman.2021.112189; Pletsch et al. (2021) DOI: 10.1590/0001-3765202120210077; Abreu et al. (2022) DOI: 10.1016/j.scitotenv.2022.157138

Author answer: thank you for the suggestion. This information was added to the text.

- Line 104: Correct the reference (Kozlowski 1974) formatting.

Author answer: corrected

- Section “3. Impact on ecosystems and wildlife”: The inclusion of these articles in this section can be interesting: de Barros et al. (2022) DOI: 10.1038/s42003-022-03937-1; Brito et al. (2017) DOI: 10.1007/s11356-017-9578-0

Author answer: added.

- Lines 142-147: Please see/cite the discussion/section "Fires and other drivers of unusual movement pattern of animals" on this paper: Ellwanger et al. (2022) DOI: 10.1590/0001-3765202220211530

Author answer: added.

- Lines 184-489: Also mention the role of biologists, NGOs, firefighters, and volunteers in fighting wildfires and rescuing animals.

Author answer: added.

- Figures: It is important to mention in the legend of the figures the source of the images used (it is not clear which images belong to the authors or which were extracted from other sources).

Author answer: the figure belongs to the authors. That information was added.

- Lines 279-208: Check the use of punctuation and commas in this sentence.

Author answer: corrected.

- Line 284: Correct the references (Venn-Watson et al 2013Black et al 2017) formatting style.

Author answer: corrected

Finally, I would like to congratulate the authors for this article dealing with an important but neglected topic.

Round 2

Reviewer 2 Report

Thank you for adding objectives statements. The purpose of this review is now more clear.

Minor English editing is needed in sections with new text.

Author Response

Authors’ answer: corrections in the text were performed as requested.